# Institutional Shareholders and Firm ESG Performance: Evidence from China

**Fang Jia [1], Yanyin Li [1], Lihong Cao [2],\*, Lintong Hu [1] and Beibei Xu [1]**

[1] School of Management, Wuhan Polytechnic University, 36 Huanhu Middle Road, Dongxihu District, Wuhan 430048, China
[2] Business School, Hunan University, Lushan South Road 2, Changsha 410082, China
\* Correspondence: caolihong@hnu.edu.cn

**Abstract:** It is a noteworthy phenomenon that institutional investors care more about the ESG performance of the firms in their portfolios in China. Exploring the role of institutional shareholders in firms' ESG performance is vital for corporate sustainable growth. Using a sample of publicly listed firms from 2013 to 2020 in China, through the OLS model, order logistic model, and tobit model, we found that firms with higher institutional ownership had better ESG performance, especially in the environmental (E) aspect. The positive effect of institutional investors on ESG performance is more pronounced in SOE firms, and firms in low pollution industries. Furthermore, mechanism tests suggest that institutional shareholders can incentivize firms to engage in ESG by affecting management change and board voting.

**Keywords:** ESG; institutional ownership; responsible investment





## 1. Introduction

The growing governments value sustainable and responsible impact in the development of the economy and encourage the financial institutions to care about the social responsibility of an investment. In recent years, the situation of climate change, energy depletion, and environment pollution have seriously affected sustainable development in the world.

In 2006, the United Nations set up the Principles and Responsible Investment (PRI) to encourage financial institution members to commit to responsible investment. Under the social demand for responsible investments, a growing number of institutions have joined in PRI and take ESG (Environment, Social Responsibility, Corporate Governance) into consideration. As shown in Figure 1, 4670 financial institutions are members of PRI at the end of 2021. Firms' ESG performance means the firms' pursuit of the maximization of social interests. ESG practice is the channel for firms to achieve sustainable development goals [1,2].

A strand of extant literature focuses on the relationship between institutional shareholders and a firm's ESG performance based on agency theory, but the conclusions are inconsistent. In the USA, the institutional shareholders improve the firm's ESG performance under the client demands and pressure of fund flows [3]. However, Chava (2014) found that institutional shareholders have a negative relationship with the corporate environmental concerns in the USA [4]. Ali et al. (2017) found that the portfolios of institutions tend to avoid firms with environment concerns [5]. Białkowski et al. (2015) found that firms with better ESG profiles tend to have investors with longer investments horizons [6]. Dyck Alexander et al. (2019) assessed the relationship between the institutional shareholders and corporate E&S performance across 41 countries. They found the relationship is affected by culture origin in different countries [7].

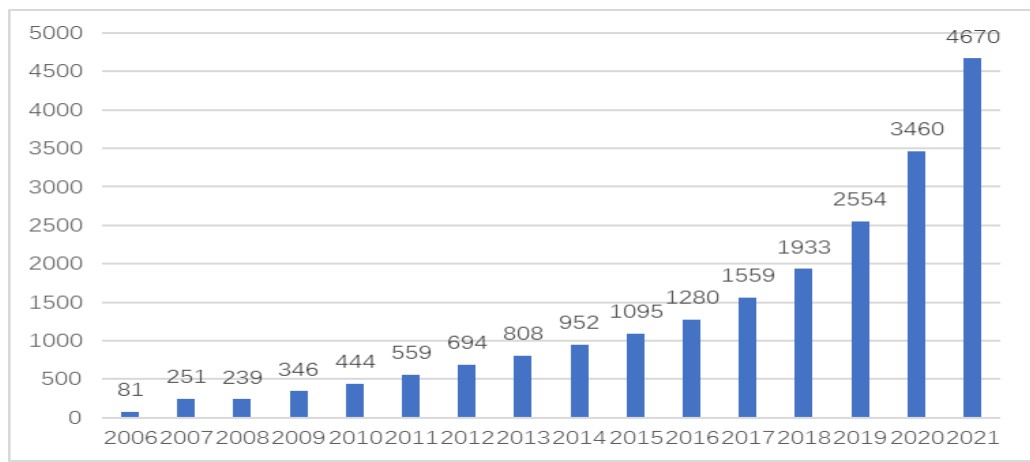

**Figure 1.** Number of members in PRI in the world from 2006–2021.

In the Chinese financial market, the institutional investments include sovereign wealth fund, mutual fund, securities, insurance fund, social security fund, annuity, privately offered fund, and QFII (Qualified foreign institutional investors). Different from some western countries, most sovereign wealth funds, insurance funds, and social security funds are guided by the Chinese government or state capital [8]. Meanwhile, the protection environment for institutional investors belonging to non-state-owned capital is relatively weak compared to some western countries [9].

The social responsible investment has become a common phenomenon in China's capital market and the institutional shareholders have become more concerned about firms' ESG performance in the portfolio. It is worthy of attention from the academic community and the industry. Although some prior research shows that institutional ownership is positively related to a firm's ESG in the USA and European financial markets, just a few studies focus on the relationship between the institution shareholders and firms' ESG performance in China. Due to the weak social network, QFII can only affect the ESG performance in non-state-owned firms significantly [10], while the state-owned institutional investors pay attention to social responsibility, such as targeted poverty reduction [11]. Allen et al. (2014) found that institutional shareholders can drive CSR performance in firms with low financial constraints [12].

Therefore, it is important to investigate whether the institutional shareholders improve the corporate ESG performance in their portfolio in China and to explore the mechanisms through which institutional investors affect a firm's ESG.

Our findings suggest that the ratio of institutional ownership has a significantly positive effect on firms' ESG performance. Furthermore, we have conducted a series of robustness tests to address endogeneity concerns and the results are consistent.

To analyze the mechanism through which institutional investors improve firms' ESG, we found two different scenarios in which the institutional investors affect firms' commitment to ESG. The first scenario is that institutional investors can influence ESG performance by actively affecting the personnel changes in management. The second scenario is that institutional investors can influence a firm's ESG performance by actively participating in board proposals.

We further investigate the impact of institutional investors on the performance of each subcategory E (Environment), S (Society), and G (Governance), respectively. While institutional shareholdings can improve all the three subcategories, the performance of the environment has been promoted the most and the improvement of corporate governance is minimal. This suggests that institutional shareholders' primary concern is for the environment rather than for governance.

At last, we investigate the moderation effect of different property rights and industries on the relationship between institutional investors and corporate ESG performance. We found that institutional investors can improve ESG performance more effectively in the SOE group and low pollution industry group.

Our study makes three important contributions to the literature on institutional investors, corporate ESG, and socially responsible investment. Firstly, our study contributes to the growing literature on corporate ESG. A lot of recent studies have focused on firm-level characteristics, shareholders characteristics, or observable managerial characteristics to explain the variation in firms' ESG [13]. With the perspective of institutional shareholders, we found the link between institutional ownership and ESG performance in the Chinese financial market through the OLS model, order logistic model, and tobit model.

Secondly, we contribute to the literature on the impact of institutional investors on corporate governance. Different from the previous literature, which mostly focused on the institutional background [10,11], we explored the channel on the improvement of ESG performance derived from institutional shareholders. Our research shows that institutional shareholders make a real effort to promote ESG performance by affecting the personnel changes in management and participating in board proposals.

Thirdly, different from the previous literature taking the ESG performance of Chinese firms as a whole [14,15], we further analyzed institutional investors' impact on the three subcategories of ESG and found that the performance of the environment has been promoted the most.

The rest of the paper is organized as follows. Section 2 introduces the background of the institutional investors on socially responsible investment in China and discusses the related literature. Section 3 describes the summary statistics and Section 4 presents the baseline results, robustness tests, the possible underlying mechanisms, and other additional tests. Section 5 concludes the paper.

## 2. Institutional Background, Related Literature, and Development

### 2.1. Institutional Background

ESG investment has experienced the stage of ethical investment since the 18th century. Since the 1990s, the Chinese government begun to pay attention to corporate social responsibility. Since 2006, with the population of ESG investment philosophy, the Chinese government and firms began to develop the ESG investment and performance.

Based on the attention to ESG performance, more listed firms begun to disclose their ESG performance reports to the stock exchanges and public investors. Panel A of Table 1 shows the distribution of ESG reports in A-share listed firms from 2013 to 2021. At the end of 2020, more than 1100 listed firms issued ESG reports. However, compared to the number of total listed firms, the proportion of listed firms with ESG reports is only 25%.

With the increasing attention to social responsibility from the Chinese government, more investment institutions are participating in the ESG investment. Panel B of Table 1 shows the distribution of ESG funds (ESG funds refer to the funds with strategies on covering the environment (E), society (S), governance (G) at the same time) and pan-ESG funds (pan-ESG funds refer to the funds with strategies on covering one or two of environment (E), society (S), governance (G)). By the end of 2021, the number of ESG funds was 69, which has almost increased eight times compared to 2017. The number of pan-ESG funds was 803, a number which has almost doubled during the past four years.

Table 1 presents the distribution of firms with ESG reports, ESG funds, and pan-ESG funds. Panel A reports the distribution of firms with ESG reports in two main stock exchanges in China from the year 2013 to 2021. Panel B reports the distribution of ESG funds and pan-ESG funds. All the data are collected from the Wind database.

**Table 1.** The distribution of firms with ESG reports, ESG funds, and pan-ESG funds.

| Panel A: distribution of firms with ESG reports | | | | |
|---|---|---|---|---|
| Year | No. of firms with ESG reports in Shenzhen Stock Exchange | No. of firms with ESG reports in Shanghai Stock Exchange | No. of firms with ESG reports | No. of firms |
| 2013 | 289 | 415 | 704 | 2489 |
| 2014 | 284 | 423 | 707 | 2612 |
| 2015 | 314 | 435 | 749 | 2827 |
| 2016 | 301 | 428 | 729 | 3050 |
| 2017 | 326 | 475 | 801 | 3485 |
| 2018 | 348 | 522 | 870 | 3582 |
| 2019 | 377 | 563 | 945 | 3773 |
| 2020 | 399 | 606 | 1005 | 4147 |
| 2021 | 428 | 702 | 1130 | 4685 |
| Panel B: distribution of ESG funds and pan-ESG funds | | | | |
| Year | No. of ESG Funds | No. of pan-ESG funds | | |
| 2017 | 9 | 423 | | |
| 2018 | 7 | 498 | | |
| 2019 | 13 | 534 | | |
| 2020 | 29 | 619 | | |
| 2021 | 69 | 803 | | |

### 2.2. Related Literature and Hypothesis

A large amount of literature discusses the factors influencing corporate ESG performance in terms of internal and external governance mechanisms.

The external stakeholders, such as the government, creditors, media, and consumers, can make contributions to the corporate ESG performance. For example, after the replacement of mayors, firms will increase social donations in order to maintain a relationship with the new government administrator [11]. Banks are more likely to lend cash to firms with good ESG profiles in order to support the ESG behavior [15] (. The media can affect public views about listed firms' ESG performance, so firms will promote the ESG performance to meet the public expectation [16]. The socially responsible corporate customers can infuse similar socially responsible business behaviors in suppliers [17].

The internal stakeholders, such as shareholders and managers, can also have a significant impact on the corporate ESG performance [18]. The personality of managers, such as their experience and family status, can influence the corporate ESG performance. The female board members will pay more attention to social issues and environment issues, and the firms will have a higher ESG performance [10].

In the USA, the ESG ratings can be improved by 9.1% if the CEO has a daughter [19]. The CEOs with military experience care about the projects with higher ESG performance [20].

In the past decade, sustainable and responsible investments (SRI) have become part of mainstream investing strategies. Some studies pay attention to the relationship between institutional shareholders and corporate ESG performance. There are two opposite views on the role of institutional shareholders on ESG performance. On the one hand, the institutional shareholders can improve the ESG performance by monitoring motivation. On the other hand, the institutional shareholders can inhibit the ESG performance by myopia motivation.

According to agency theory, institutional investors, especially mutual funds and pension funds, are active and effective monitors for internal corporate governance [21]. Due to their strong ability to collect information, institutional shareholders can effectively monitor corporate governance and influence decision making through the advantages of resources and expertise. Moreover, investment horizons of institutional investors can affect

monitoring motivation, which in turn affects various decisions of firms [22,23] such as ESG activities.

There are three main motives for institutional investors to care about the ESG performance. Firstly, the beliefs of institutional investors in social responsibility have important impacts on the involvement in SRI [24]. The questionnaires released by Zwaan (2015) show the sovereign wealth funds, pension funds, and NGO funds are more interested in firms with higher ESG performance [25].

Secondly, the public's preference for ESG will encourage institutional investors to invest in firms with high ESG levels [17]. More individual investors will consider environmental and social impacts when making investment decisions. They are willing to invest in socially responsible firms with a high financial premium [26]. Social preferences are more effective than financial motives in the explanation of SRI [7].

Thirdly, institutional investors can use SRI in portfolios for risk management. They believe that environmental risks have financial implications for their portfolio. Especially in long-term investments, it is a better approach to participate in environmental risk management rather than divestment [27]. Jose Azar et al. (2021) found that the Big Three (i.e., BlackRock, Vanguard, and State Street Global Advisors) focus their portfolios on the firms with high $CO_2$ emissions and make efforts to reduce the $CO_2$ emissions from these firms [28]. Dyck et al. (2019) found that companies with better ESG performance tend to have investors with longer investment horizons [7].

However, according to stakeholder theory, some prior studies believe the institutional shareholders can inhibit the ESG performance by myopia. The institutional shareholders' concerns about short-term interests will lead the firm to pursue short-term profits [29,30]. Although the improvement of ESG performance can increase firm value and shareholder wealth in the long term [31]. The practice of ESG, such as environmental protection and employee welfare, will reduce the earnings and cash in the current term by adding additional expenses and costs [16]. Furthermore, the disclosure of the improvement in firms' ESG performance would reduce the market value [32]. Therefore, the improvement of ESG performance is at the expense of shareholders' wealth. Based on this analysis, institutional shareholders will have the incentive to reduce the firms' ESG performance.

In the total, based on the prior studies, this paper examines the relationship between institutional investors and ESG performance of A-shares listed firms in China. We propose the following competitive hypotheses:

**H1. Institutional investors can improve the ESG performance in A-shares listed firms**.

**H2. Institutional investors can inhibit the ESG performance in A-shares listed firms**.

### 3. Data and Variables

*3.1. Data*

We obtained the data of institutional shareholders from the China Stock Market & Accounting Research (CSMAR) Database. CSMAR collects the number of shares held by institutional shareholders from the firms' annual reports. We further obtained institutional shareholders' information and financial data of firms from CSMAR.

We obtained the ESG data of listed firms from the Wind Database. As there has been increasing social attention to ESG, many ESG rating systems have emerged in China, such as social value investment alliance rating, SynTao green finance rating, and Harvest ESG rating. However, compared with the Wind ESG rating, other systems have a lower update frequency and narrower coverage of A-share listed firms. The Wind ESG rating system is built on mainstream ESG system frameworks in developed countries. In addition, the Wind ESG rating system adds many indicators which reflect Chinese ESG characteristics such as public opinion, poverty alleviation, and CSRC (China Securities Regulatory Commission) punishment. Furthermore, the Wind ESG rating covers more than 20,000 data sources including corporate annual reports, government announcements, and media re-

ports. Therefore, we used the data of Wind ESG rating to measure the ESG performance of firms.

Our initial sample included the panel data of all A-share listed firms from 2013 to 2020. During the National Securities Dealers Innovation Conference held in 2012, the Chinese government eased the restriction on the proportion of shares held by securities. Since the institutional investors became more active during the secondary market and the shares held by institutional investors have grown significantly. Therefore, our sample starts in 2013. We then screened the initial sample as follows: (1) The financial industry firms are deleted because their financial statements are specifically different from other industry firms; (2) ST firms are deleted; (3) Observations with missing data are deleted.

Our final sample consists of 16,810 firm-year observations for the non-financial firms from 2013 to 2020.

### *3.2. Variables*

The Wind ESG rating system includes three levels: (1) the first level indicators are environmental (E), social (S), and corporate governance (G); (2) the second level indicators are 27 classified indicators under ESG issues; (3) the third level indicators are more than 300 classified indicators under the second level indicators. The Wind ESG rating results are divided into two evaluation methods: ESG rating level and ESG scoring level. The ESG rating level is divided in nine grades of AAA to C. Therefore, according to the nine rating levels, we assigned AAA–C as 1–9 in turn. We noted ESGlevel = 9 when the rating level is AAA, ESGlevel = 1 when the rating level is C, and so on. The scoring rating method is presented in the form of a comprehensive score, which is recorded as ESGscore. However, the data of the ESGScore are published from 2017, so we can only analyze 7477 firm-year observations from 2017 to 2020.

According to Dyck et al., (2019) [7], we used the percentage of the total ownership of institutional investors in the total ownership as the proportion of institutional investors.

We also controlled other firm characteristic indicators, such as firms' age, book-to-market value, cash holding level, growth ability, ownership concentration, ratio of independent directors, and so on. We winsorized the data at 1% and 99% levels.

Table 2 provides detailed descriptions and definitions of all the variables used in this paper.

**Table 2.** Variable Definition.

| Variable | Definitions |
|---|---|
| ESGlevel | Wind ESG rating on 1 to 9 from low to high |
| ESGscore | Wind ESG score |
| INS | Number of shares held by institutional investors at the end of the year/ number of outstanding shares at the end of the year |
| Lev | Total liabilities/total assets |
| Age | The natural logarithm of the company's age |
| Cash | Cash and cash equivalents/total assets |
| BM | Book value/market value |
| Growth | Growth rate of sales |
| Top1 | Shareholding ratio of the largest shareholder |
| RID | Number of independent directors/number of directors |
| DRA | Dummy variable equal to one if the CEO is also chairman of the board in year t, and zero otherwise |

## 4. Empirical Results and Analysis

### *4.1. Empirical Model*

Following Dyck et al. (2019) [7], we estimate the following baseline model to investigate the relationship between shares held by institutional shareholders and ESG performance:

$$ESGlevel_{i,t}/ESGleveScore_{i,t} = \alpha_0 + \alpha_1 INS_{i,t} + \alpha_j Controls_{i,t} + Year_t + Ind_i + \varepsilon_{i,t} \quad (1)$$

In Equation (1), $i$ represents the individual firm, $t$ represents the year, and $Controls_{i,t}$ represent all the control variables involved in this article which are divided into firm characteristics, whilst Yeart and Indi capture the industry and year fixed effects.

The table presents the summary characteristics (mean) for the sample firms. The sample is comprised of 16,810 firm-year observations during the 2013 to 2020 period. The appendix provides detailed descriptions of the variables.

Table 3 shows the descriptive statistics for all variables in the present study. The final sample consisted of 16,810 firm-year observations. The statistical results show that the average value of the ESGlevel is 6.526, the max value is 9, and min value is 1. The average value of the ESGscore is 6.099, the max value is 8.1, and the min value is 4.73. According to the distribution of ESGscore and ESGlevel, we can obtain a large gap in ESG performance among our observations.

**Table 3.** Summary Statistics.

| Variable | N | Max | Min | Mean | p50 |
|---|---|---|---|---|---|
| ESGlevel | 16,810 | 9 | 1 | 6.526 | 6 |
| ESGscore | 7477 | 8.100 | 4.730 | 6.099 | 6.020 |
| ins | 16,810 | 83.78 | 0.491 | 39.87 | 41.29 |
| age | 16,810 | 3.526 | 2.398 | 3.032 | 3.045 |
| cash | 16,810 | 0.536 | 0.0130 | 0.149 | 0.120 |
| lev | 16,810 | 0.844 | 0.0770 | 0.426 | 0.418 |
| roa | 16,810 | 0.177 | −0.145 | 0.0430 | 0.0390 |
| size | 16,810 | 25.68 | 20.22 | 22.29 | 22.14 |
| bm | 16,810 | 1.121 | 0.153 | 0.618 | 0.610 |
| growth | 16,810 | 1.367 | −0.434 | 0.144 | 0.101 |
| top1 | 16,810 | 68.00 | 10.52 | 34.35 | 32.55 |
| rid | 16,810 | 0.500 | 0.333 | 0.373 | 0.364 |

The average value of the ratio of institutional ownership is 39.87%, whilst the max value is 83.78%. This shows that institutional investors can have an important influence on the company's decision making in some firms.

Table 4 shows the correlation analysis for all variables in the present study to evaluate the rationality of variable selection.

**Table 4.** Correlation analysis.

| | ESGlevel | ESGscore | Ins | Age | Cash | Lev | Roa | Size | Bm | Growth | Top1 | Rid |
|---|---|---|---|---|---|---|---|---|---|---|---|---|
| ESGlevel | 1 | | | | | | | | | | | |
| ESGscore | 0.344 *** | 1 | | | | | | | | | | |
| ins | 0.249 *** | 0.107 *** | 1 | | | | | | | | | |
| age | 0.116 *** | 0.019 * | 0.160 *** | 1 | | | | | | | | |
| cash | 0.058 *** | 0.067 *** | 0.00300 | −0.044 *** | 1 | | | | | | | |
| lev | 0.102 *** | 0.00100 | 0.211 *** | 0.171 *** | −0.343 *** | 1 | | | | | | |
| roa | 0.111 *** | 0.065 *** | 0.051 *** | −0.085 *** | 0.264 *** | −0.374 *** | 1 | | | | | |
| size | 0.339 *** | 0.187 *** | 0.434 *** | 0.170 *** | −0.204 *** | 0.556 *** | −0.064 *** | 1 | | | | |
| bm | 0.126 *** | 0.00100 | 0.050 *** | 0.124 *** | −0.248 *** | 0.435 *** | −0.264 *** | 0.573 *** | 1 | | | |
| growth | −0.00600 | −0.0150 | −0.036 *** | −0.090 *** | 0.00500 | 0.0120 | 0.269 *** | 0.015 ** | −0.069 *** | 1 | | |
| top1 | 0.126 *** | 0.029 ** | 0.381 *** | −0.015 * | 0.058 *** | 0.055 *** | 0.106 *** | 0.157 *** | 0.094 *** | −0.030 *** | 1 | |
| rid | −0.016 ** | −0.00500 | −0.073 *** | −0.038 *** | 0.025 *** | −0.024 *** | −0.00700 | −0.034 *** | −0.042 *** | 0.015 ** | 0.022 *** | 1 |

***, **, and * indicate the significance of variables at the 1%, 5%, and 10% levels, respectively.

Table 4 shows the correlation analysis for all variables in the present study to evaluate the rationality of variable selection. We can observe that the correlation coefficient between institutional investors and ESG performance is significant at the level of 1%, which preliminarily shows that there is a positive correlation between institutional investors and company ESG performance, which is in line with the hypothesis.

### 4.2. Baseline Regression Results

We examined the impact of the ratio of institutional ownership on firm ESG performance. Table 5 shows the estimation results of Equation (1) by using OLS regressions.

**Table 5.** Institutional ownership and ESG performance.

|  | (1) | (2) | (3) | (4) | (5) | (6) |
|---|---|---|---|---|---|---|
|  | ESGlevel | ESGscore | ESGlevel | ESGscore | ESGlevel | ESGscore |
| ins | 0.0126 *** | 0.0032 *** | 0.0107 *** | 0.0032 *** | 0.0101 *** | 0.0036 *** |
|  | (0.0004) | (0.0003) | (0.0004) | (0.0004) | (0.0004) | (0.0004) |
| age |  |  | 0.2953 *** | −0.0031 | 0.1349 *** | 0.0583 * |
|  |  |  | (0.0348) | (0.0313) | (0.0346) | (0.0314) |
| cash |  |  | 1.0305 *** | 0.4403 *** | 0.6891 *** | 0.3383 *** |
|  |  |  | (0.0846) | (0.0810) | (0.0844) | (0.0819) |
| lev |  |  | 0.1364 ** | −0.0114 | −0.1115 ** | 0.0649 |
|  |  |  | (0.0521) | (0.0489) | (0.0534) | (0.0505) |
| bm |  |  | 0.5471 *** | 0.0345 | 0.6480 *** | 0.1457 *** |
|  |  |  | (0.0398) | (0.0375) | (0.0439) | (0.0396) |
| growth |  |  | 0.0705 ** | −0.0312 | 0.0523 * | −0.0144 |
|  |  |  | (0.0293) | (0.0299) | (0.0289) | (0.0299) |
| top1 |  |  | 0.0023 *** | −0.0009 | 0.0018 ** | 0 |
|  |  |  | (0.0007) | (0.0006) | (0.0007) | (0.0006) |
| rid |  |  | 0.2218 | 0.1004 | 0.3268 * | 0.0734 |
|  |  |  | (0.1776) | (0.1659) | (0.1725) | (0.1634) |
| dra |  |  | −0.0998 *** | −0.0303 * | −0.0929 *** | −0.0316 * |
|  |  |  | (0.0198) | (0.0182) | (0.0192) | (0.0180) |
| Year | No | No | No | No | Yes | Yes |
| Industry | No | No | No | No | Yes | Yes |
| _cons | 6.0232 *** | 5.9735 *** | 4.5101 *** | 5.9038 *** | 5.0400 *** | 5.3968 *** |
|  | (0.0174) | (0.0157) | (0.128) | (0.1155) | (0.1432) | (0.1335) |
| $N$ | 16,810 | 7477 | 16,810 | 7474 | 16,810 | 7474 |
| $R^2$ | 0.062 | 0.011 | 0.089 | 0.016 | 0.15 | 0.053 |

The *t*-statistics are presented in the parenthesis and superscripts ***, **, and * denote statistical significance at the 1%, 5%, and 10% levels, respectively.

This table reports pooled regressions of the ESG performance variables on the percentage of shares held by institutional shareholders and other control variables. All control variables are defined in Table 2. In column (1) and (2), there are no control variables. In column (3) and (4), all variables are controlled in the Equation (1). In column (5) and (6), the regression includes year and industry fixed effects. The robust standard errors are clustered by firms.

The coefficients of ESGlevel and ESGscore in column (1) and (2) are positive and significant at the 1% level, suggesting that a higher ratio of institutional ownership is associated with a higher level of ESG performance. Furthermore, after we controlled the variables of firm characters, the coefficients of ESGlevel and ESGscore in column (3) and (4) are still positive and significant at the 1% level. Lastly, after we controlled the industry and year fixed effects, the coefficients of ESGlevel and ESGscore in column (5) and (6) are 0.0101 and 0.0036, respectively, and significant at the 1% level. Overall, according to the results in Table 5, we can conclude that the ratio of institutional ownership is significantly positively related to the firm's ESG performance.

### 4.3. Robustness Checks

We also performed a series of additional tests to ensure that the significant positive relationship between institutional ownership and a firm's ESG performance is robust to model specifications, variable definitions, and lag period.

The table report pooled regressions of the ESG performance on the percentage held by institutional shareholders and other control variables by using different model specifications, variable definitions, and lag period. Panel A pooled the regression in order logistic model in ESGlevel and tobit model in ESGscore. Panel B assigned the dependent variable of ESGlevel in three levels. Panel C lags the covariates by three years. The robust standard errors are clustered by firms. The t-statistics are presented in the parentheses and superscripts ***, **, and * denote statistical significance at the 1%, 5%, and 10% levels, respectively.

Panel A of Table 6 shows the regression in order logistic model in ESGlevel and tobit model in ESGscore. Ordinal logistic regression is suitable for ordinal variables which have rank or degree difference. In this study, the dependent variables of ESGlevel are assigned in ordinal value. Therefore, we used the order logistic model to test the relationship between ESGlevel and institutional ownership. Panel A presents the statistic results between ESGlevel and institutional ownership in order logistic model without year and industry fixed effects in column (1) and with year and industry fixed effects in column (2), respectively. The tobit model refers to a type of model in which the dependent variable is roughly continuously distributed on the positive value, but contains a part of the observations with a positive probability value of 0. The dependent variables, ESGscore, are assigned to be larger than 0, so we chose the tobit model for the dependent variables ESGscore. In column (3) and column (4), we represent the statistic results between ESGscore and institutional ownership in the tobit model without year and industry fixed effects and with year and industry fixed effects, respectively. In panel A, the ESG performance in a different regression model is still significantly related to institutional ownership at the 1% level.

**Table 6.** Robustness test.

| | Panel A: order logistic model in ESGlevel and tobit model in ESGscore | | | |
| --- | --- | --- | --- | --- |
| | (1)<br>ESGlevel | (2)<br>ESGscore | (3)<br>ESGlevel | (4)<br>ESGscore |
| ins | 0.0106 *** | 0.0032 *** | 0.0178 *** | 0.0036 *** |
| | (0.0004) | (0.0004) | (0.0007) | (0.0004) |
| age | 0.2983 *** | −0.0017 | 0.2039 *** | 0.049 |
| | (0.0348) | (0.0313) | (0.0592) | (0.0308) |
| cash | 1.0144 *** | 0.4286 *** | 1.0842 *** | 0.3117 *** |
| | (0.0848) | (0.0812) | (0.1418) | (0.0815) |
| lev | 0.1360 ** | −0.0258 | −0.2110 ** | 0.0612 |
| | (0.0521) | (0.0492) | (0.0937) | (0.0516) |
| bm | 0.5404 *** | 0.0475 | 1.1857 *** | 0.1542 *** |
| | (0.0399) | (0.0379) | (0.0779) | (0.0403) |
| growth | 0.0691 ** | −0.0247 | 0.0315 | −0.0166 |
| | (0.0293) | (0.0300) | (0.0467) | (0.0286) |
| top1 | 0.0023 *** | −0.0008 | 0.0040 *** | 0.0002 |
| | (0.0007) | (0.0006) | (0.0012) | (0.0006) |
| rid | 0.2139 | 0.1008 | 0.7067 ** | 0.0898 |
| | (0.1776) | (0.1658) | (0.2898) | (0.1610) |
| dra | −0.1006 *** | −0.0301 * | −0.1672 *** | −0.0355 * |
| | (0.0198) | (0.0182) | (0.0335) | (0.0184) |
| Year | No | No | Yes | Yes |
| Industry | No | No | Yes | Yes |
| _cons | 4.4865 *** | 5.8734 *** | 5.0380 *** | 5.4067 *** |
| | (0.1282) | (0.1162) | (0.1471) | (0.1326) |
| $N$ | 16,810 | 7477 | 16,810 | 7477 |
| $R^2$ | 0.09 | 0.017 | 0.151 | 0.054 |

**Table 6.** *Cont.*

| Panel B: the dependent variable of ESGlevel is assigned in three levels | | |
|---|---|---|
| | (1) ESGlevel2 | (2) ESGlevel2 |
| ins | 0.0040 *** | 0.0037 *** |
| | (0.0002) | (0.0002) |
| age | 0.1189 *** | 0.0489 ** |
| | (0.0153) | (0.0152) |
| cash | 0.5051 *** | 0.3061 *** |
| | (0.0372) | (0.0369) |
| lev | 0.1183 *** | −0.0225 |
| | (0.0235) | (0.0240) |
| bm | 0.2078 *** | 0.2308 *** |
| | (0.0180) | (0.0197) |
| growth | 0.0053 | 0.0012 |
| | (0.0123) | (0.0121) |
| top1 | 0.0008 ** | 0.0008 ** |
| | (0.0003) | (0.0003) |
| rid | 0.0307 | 0.0431 |
| | (0.0772) | (0.0745) |
| dra | −0.0407 *** | −0.0393 *** |
| | (0.0090) | (0.0087) |
| Year | No | Yes |
| Industry | No | Yes |
| _cons | 1.6288 *** | 1.9465 *** |
| | (0.0563) | (0.0631) |
| *N* | 16,810 | 16,810 |
| $R^2$ | 0.072 | 0.143 |

| Panel C: lag the covariates by one year and two years | | | | |
|---|---|---|---|---|
| | (1) ESGlevel_lag1 | (2) ESGscore_lag1 | (3) ESGlevel_lag2 | (4) ESGscore_lag2 |
| ins | 0.0096 *** | 0.0032 *** | 0.0083 *** | 0.0029 *** |
| | (0.0005) | (0.0005) | (0.0006) | (0.0009) |
| age | 0.1275 ** | 0.1023 ** | 0.1244 ** | 0.0926 |
| | (0.0388) | (0.0417) | (0.0441) | (0.0653) |
| cash | 0.6805 *** | 0.3235 ** | 0.6501 *** | 0.1201 |
| | (0.0974) | (0.1126) | (0.1119) | (0.1723) |
| lev | −0.0314 | 0.077 | 0.0321 | 0.1191 |
| | (0.0612) | (0.0701) | (0.0677) | (0.1068) |
| bm | 0.6410 *** | 0.1918 *** | 0.6218 *** | 0.1688 ** |
| | (0.0496) | (0.0530) | (0.0536) | (0.0794) |
| growth | −0.0558 * | 0.0319 | (0.0555) | (0.0549) |
| | (0.0306) | (0.0394) | (0.0345) | (0.0597) |
| top1 | 0.0006 | −0.0003 | 0.0002 | −0.0006 |
| | (0.0008) | (0.0009) | (0.0009) | (0.0014) |
| rid | 0.4152 ** | 0.0517 | 0.4077 * | 0.228 |
| | (0.1877) | (0.2159) | (0.2084) | (0.3331) |
| dra | −0.0719 ** | −0.017 | −0.0927 *** | −0.0386 |
| | (0.0221) | (0.0249) | (0.0248) | (0.0389) |
| Year | Yes | Yes | Yes | Yes |
| Industry | Yes | Yes | Yes | Yes |
| _cons | 5.2324 *** | 5.3005 *** | 5.3794 *** | 5.3782 *** |
| | −0.1576 | −0.177 | −0.1755 | −0.2689 |
| *N* | 12,501 | 4266 | 9946 | 1791 |
| $R^2$ | 0.147 | 0.049 | 0.135 | 0.059 |

The *t*-statistics are presented in the parenthesis and superscripts ***, **, and * denote statistical significance at the 1%, 5%, and 10% levels, respectively.

Secondly, we changed the method of valuation on dependent variables ESGlevel. In panel B, according to the categories of ESG rating (class A, B, C), we assigned ESGlevel in 1, 2, 3. When the rating was class A, ESGlevel2 = 3; When rated as class B, ESGlevel2 = 2; When rated as class C, ESGlevel3 = 1. We presented the statistic results between ESGlevel and institutional ownership without year and industry fixed effects in column (1) and with year and industry fixed effects in column (2), respectively. We found that the positive relation between the ESGlevel and institutional ownership is still robust.

Moreover, ESG activities are long-term projects, hence the institutional ownership may not affect the corporate ESG performance intermediately and the effect may work two years or more into the future. Therefore, the dependent variable, which is the corporate ESG performance in the present year, may not be enough to support our assumption. To ensure our baseline results are robust, we investigated whether the institutional ownership affects ESG performance two years ahead. In panel C, we can see that institutional ownership is still positively related to two-year-ahead ESG performance at the 1% level.

### 4.4. Mechanism of Institutional Ownership Impact on ESG Performance

Although we found that institutional ownership has a positive effect on ESG performance, the mechanism through which institutional ownership improves ESG performance is still unclear. We propose two different ways in which institutional investors influence the firm's ESG performance.

The first scenario is that institutional investors can improve the ESG performance by actively affecting the personnel changes in management. Improvement of ESG is a long-term development strategy, which is related to the long-term development of the company and the long-term external impact on the social environment. However, the management needs short-term financial benefits to obtain compensation returns or a good reputation [33]. Institutional investors can use the voice brought by their shareholding to promote ESG by actively participating in corporate governance [34], and influencing management changes in a way for institutional investors to get involved [18]. Therefore, management changes play a mediating effect between institutional investors and corporate ESG performance.

The second scenario is that institutional investors can affect a company's ESG performance by actively participating in board voting. For example, one could make ESG proposals on the board of directors, actively elect people who are willing to promote the development of ESG to the board of directors, approve the proposals related to the ESG strategy of the company, reject the proposals that will reduce the performance of ESG of the company, and so on. Therefore, active exercise of shareholder voting rights is an important way for institutional investors to perform and supervise the company [26] Therefore, board consent plays another mediating effect between institutional investors and corporate ESG performance.

We denote variable $Change_{i,t}$ as the number of management changes in firm $i$ in year $t$, variable $Proposal_{i,t}$ as the number of board consents in firm $i$ in year $t$. We use the following model to test the mediating effect in two scenarios:

$$ESGlevel_{i,t}/ESGleveScore_{i,t} = \alpha_0 + \alpha_1 INS_{i,t} + \alpha j Controls_{i,t} + Year_i + Ind_j + \varepsilon_{i,t} \quad (2)$$

$$Change_{i,t}/Proposal_{i,t} = \beta_0 + \beta_1 INS_{i,t} + \beta_j Controls_{i,t} + Year_i + Ind_j + \varepsilon_{i,t} \quad (3)$$

$$ESGlevel_{i,t}/ESGleveScore_{i,t} = \gamma_0 + \gamma_1 INS_{i,t} + \gamma_2 Change_{i,t}/Prososal_{i,t} + \gamma_j Controls_{i,t} + Year_i + Ind_j + \varepsilon_{i,t} \quad (4)$$

In panel A of Table 7, the results of management change as a mediating variable are reported. In column (1), the influence coefficients of institutional investors' shareholding ratio on ESG score and ESG rating performance are 0.887 and 0.659, respectively, which are significant at the 1% level. In the further test, column (2) shows that the mediating variable management Change has a significant positive correlation with the explanatory variable at the 1% level, with a coefficient of 0.125, indicating that the larger the shareholding

ratio of institutional investors, the more frequent the management changes, and there is a mediating effect. From the regression results in column (3), it can be found that the direct effect of institutional investors is significant at the 10% level with a coefficient of 0.375, while the regression coefficient of the intermediary variable management change is significant at the 1% level, indicating that there is a partial mediation effect. That is, when other conditions remain unchanged, institutional investors can improve ESG performance by adjusting management, and the mechanism test is verified.

**Table 7.** Mechanism test.

| | Panel A: Management changes | | | | |
|---|---|---|---|---|---|
| | (1) ESGlevel | (2) ESGscore | (3) Change | (4) ESGlevel | (5) ESGscore |
| ins | 0.887 *** | 0.659 ** | 0.125 *** | 0.385 * | 0.297 * |
| | (0.2730) | (0.2810) | (0.0060) | (0.2120) | (0.1920) |
| Change | | | | 2.315 *** | 1.915 *** |
| | | | | (0.2660) | (0.3762) |
| age | −0.101 | 0.109 | 0.003 *** | 0.127 *** | 0.113 *** |
| | (0.0750) | (0.0290) | (0.0010) | (0.0340) | (0.0410) |
| cash | 0.38 * | 0.351 *** | −0.005 ** | 0.624 *** | 0.529 *** |
| | (0.2000) | (0.0730) | (0.0020) | (0.0820) | (0.0910) |
| lev | 0.216 * | 0.372 | 0.0279 *** | −0.0478 | −0.0436 |
| | (0.1260) | (0.0420) | (0.0020) | (0.0540) | (0.0610) |
| bm | 0.275 *** | 0.183 *** | −0.011 *** | 0.621 *** | 0.597 *** |
| | (0.0990) | (0.0500) | (0.0010) | (0.0440) | (0.0510) |
| growth | −0.11 | −0.023 | −0.005 *** | 0.027 | 0.031 |
| | (0.0700) | (0.0220) | (0.0010) | (0.0270) | (0.0180) |
| top1 | 0.001 | 0.001 | −0.0001 *** | 0.001 ** | 0.001 ** |
| | (0.0020) | (0.0020) | 0.0000 | (0.0010) | 0.0000 |
| rid | 0.096 | 0.091 | 0.004 | 0.381 ** | 0.401 ** |
| | (0.3950) | (0.1410) | (0.0050) | (0.1650) | (0.1910) |
| dra | −0.097 ** | −0.034 ** | −0.001 * | −0.087 | −0.102 |
| | (0.0450) | (0.0210) | (0.0010) | (0.0200) | (0.0290) |
| Year | Yes | Yes | Yes | Yes | Yes |
| Ind | Yes | Yes | Yes | Yes | Yes |
| _cons | 5.941 *** | 5.47 *** | 5.023 *** | 5.021 *** | 4.817 *** |
| | (0.0590) | (0.1520) | (0.0040) | (0.1420) | (0.1510) |
| *N* | 16,810 | 7477 | 16,810 | 16,810 | 7477 |
| *R²* | 0.033 | 0.049 | 0.087 | 0.037 | 0.058 |
| | Panel B: Proposal from Board | | | | |
| | (1) ESGlevel | (2) ESGscore | (3) Proposal | (4) ESGlevel | (5) ESGscore |
| ins | 0.887 *** | 0.659 ** | 0.019 ** | 0.417 *** | 0.140 *** |
| | (0.2730) | (0.2810) | (0.0080) | (0.3750) | (0.0030) |
| Proposal | | | | 3.119 *** | 2.793 *** |
| | | | | (0.5960) | (0.3720) |
| age | −0.101 | 0.109 | 0.085 | 0.092 | 0.113 |
| | (0.0750) | (0.0290) | (0.0150) | (0.0530) | (0.0420) |
| cash | 0.38 * | 0.351 *** | −0.002 * | 0.047 *** | 0.721 *** |
| | (0.2000) | (0.0730) | (0.0010) | (0.0030) | (0.1110) |
| lev | 0.216 * | 0.372 | 0.0315 *** | −0.009 | −0.151 |
| | (0.1260) | (0.0420) | (0.0050) | (0.0210) | (0.0610) |
| bm | 0.275 *** | 0.183 *** | −0.016 *** | 0.112 *** | 0.652 *** |
| | (0.0990) | (0.0500) | (0.0010) | (0.0290) | (0.0590) |

**Table 7.** *Cont.*

| | | | | | |
|---|---|---|---|---|---|
| growth | −0.11 | −0.023 | −0.011 *** | 0.021 | 0.0243 |
| | (0.0700) | (0.0220) | (0.0020) | (0.0250) | (0.0370) |
| top1 | 0.001 | 0.001 | −0.0001 *** | 0.001 ** | 0.0036 *** |
| | (0.0020) | (0.0020) | 0.0000 | (0.0010) | (0.0009) |
| rid | 0.096 | 0.091 | 0.096 | 0.003 ** | 0.496 ** |
| | (0.3950) | (0.1410) | (0.4170) | (0.0010) | (0.2320) |
| dra | −0.097 ** | −0.034 ** | −0.049 *** | −0.048 | −0.0897 *** |
| | (0.0450) | (0.0210) | (0.0030) | (0.0290) | (0.0287) |
| Year | Yes | Yes | Yes | Yes | Yes |
| Ind | Yes | Yes | Yes | Yes | Yes |
| _cons | 5.941 *** | 5.47 *** | 6.011 *** | 5.721 *** | 5.143 *** |
| | (0.0590) | (0.1520) | (0.0720) | (0.1730) | (0.1420) |
| $N$ | 16,810 | 7477 | 16,810 | 16,810 | 7477 |
| $R^2$ | 0.033 | 0.049 | 0.042 | 0.045 | 0.051 |

The *t*-statistics are presented in the parentheses and the superscripts ***, **, and * denote statistical significance at the 1%, 5%, and 10% levels, respectively.

In panel B of Table 7, the results of board proposal as a mediating variable are reported. In column (2), it shows that the mediating variable, board resolution, has a significant positive correlation with the explanatory variable at the 1% level, indicating that the larger the shareholding ratio of institutional investors, the larger the number of board resolutions, and there is a mediating effect. From the regression results in column (3), it is observed that the direct effect of institutional investors is significant at the 10% level, while the regression coefficient of the intermediary variable, the number of board decisions, is significant at the 1% level, indicating that there is a partial mediation effect. That is, if other conditions remain unchanged, institutional investors can improve ESG performance by actively participating in the board of directors.

This table reports the mediating effect results on management changes and proposals from boards. Variable $Change_{i,t}$ as the number of management changes in firm i in year t, variable $Proposal_{i,t}$ as the number of board consents in firm i in year t. Each regression includes year and firm fixed effects.

*4.5. Additional Test*

We found that the institutional ownership can positively affect the ESG performance. Furthermore, we conducted a series of tests to analyze the deeper relationship in the subcategories of ESG, SOE and non-SOE firms, and high-pollution and low-pollution industry firms.

Firstly, to better understand what aspect of ESG issues are most affected by institutional shareholders, we extended the baseline specification to separately study the effect of shareholders on the three different dimensions of ESG activities. In panel A of Table 8, we can see the different relationships between E(environment), S(Society), G(Governance), and institutional ownership in the OLS regression of column (1), (2), (3) and the tobit regression in column (4), (5), (6). Both in OLS regression and tobit regression models, the coefficients of E are the largest, then followed by S at a 1% significance level. The coefficients of G are the smallest at a 10% significance level. This demonstrates that institutional investors pay the most attention to the environmental protection performance of companies, and they will actively help companies improve their environmental management, increase environmental information disclosure, and reduce negative environmental events. Some of these changes include reducing $CO_2$ emissions, increasing the use of renewable energy, community contribution, product liability, and so on.

**Table 8.** Additional test.

| Panel A: Institutional ownership and subcategory of ESG (E, S, G) in OLS and tobit regression | | | | | | |
|---|---|---|---|---|---|---|
| | (1)<br>E | (2)<br>S | (3)<br>G | (4)<br>E | (5)<br>S | (6)<br>G |
| ins | 0.0122 *** | 0.0074 *** | 0.0007 * | 0.0122 *** | 0.0074 *** | 0.0007 * |
| | (0.0009) | (0.0010) | (0.0004) | (0.0009) | (0.0010) | (0.0004) |
| age | 0.3172 *** | −0.1003 | 0.0755 ** | 0.3172 *** | −0.1003 | 0.0755 ** |
| | (0.0711) | (0.0754) | (0.0299) | (0.0709) | (0.0753) | (0.0298) |
| cash | −0.1083 | 0.3817 * | 0.3264 *** | −0.1083 | 0.3817 * | 0.3264 *** |
| | (0.1882) | (0.1997) | (0.0792) | (0.1878) | (0.1994) | (0.0790) |
| lev | 0.5645 *** | 0.2151 * | −0.0098 | 0.5645 *** | 0.2151 * | −0.0098 |
| | (0.1191) | (0.1264) | (0.0501) | (0.1189) | (0.1262) | (0.0500) |
| bm | 0.6545 *** | 0.2769 ** | 0.1671 *** | 0.6545 *** | 0.2769 ** | 0.1671 *** |
| | (0.0931) | (0.0989) | (0.0392) | (0.0930) | (0.0987) | (0.0391) |
| growth | −0.0921 | −0.105 | −0.0036 | −0.0921 | −0.105 | −0.0036 |
| | (0.0661) | (0.0702) | (0.0278) | (0.0660) | (0.0701) | (0.0278) |
| top1 | −0.0015 | 0.0013 | 0.0006 | −0.0015 | 0.0013 | 0.0006 |
| | (0.0015) | (0.0016) | (0.0006) | (0.0015) | (0.0016) | (0.0006) |
| rid | −0.1477 | 0.0875 | 0.4359 ** | −0.1477 | 0.0875 | 0.4359 ** |
| | (0.3717) | (0.3946) | (0.1564) | (0.3710) | (0.3938) | (0.1561) |
| dra | −0.1102 ** | −0.0959 ** | −0.0366 ** | −0.1102 ** | −0.0959 ** | −0.0366 ** |
| | (0.0425) | (0.0451) | (0.0179) | (0.0424) | (0.0450) | (0.0178) |
| Year | Yes | Yes | Yes | Yes | Yes | Yes |
| Industry | Yes | Yes | Yes | Yes | Yes | Yes |
| _cons | −0.5444 * | 3.6487 *** | 6.2968 *** | −0.5444 * | 3.6487 *** | 6.2968 *** |
| | (0.3062) | (0.3251) | (0.1289) | (0.3057) | (0.3245) | (0.1286) |
| $N$ | 7474 | 7474 | 7474 | 7474 | 7474 | 7474 |
| $R^2$ | 0.086 | 0.06 | 0.053 | 0.082 | 0.057 | 0.051 |

| Panel B: SOE and non-SOE group | | | | |
|---|---|---|---|---|
| | (1)<br>ESGlevel | (2)<br>ESGscore | (3)<br>ESGlevel | (4)<br>ESGscore |
| | SOE | SOE | Non-SOE | Non-SOE |
| ins | 0.0138 *** | 0.0074 *** | 0.0056 *** | 0.0016 *** |
| | (0.0008) | (0.0009) | (0.0005) | (0.0005) |
| age | 0.0051 | 0.084 | 0.0382 | 0.0202 |
| | (0.0617) | (0.0626) | (0.0397) | (0.0355) |
| cash | 0.3867 ** | −0.2771 * | 0.4982 *** | 0.4621 *** |
| | (0.1438) | (0.1564) | (0.0976) | (0.0955) |
| lev | −0.2589 ** | −0.1211 | −0.2149 ** | 0.1032 |
| | (0.0867) | (0.0918) | (0.0670) | (0.0630) |
| bm | 0.7857 *** | 0.3005 *** | 0.2432 *** | −0.0265 |
| | (0.0684) | (0.0700) | (0.0572) | (0.0508) |
| growth | 0.0243 | −0.0656 | 0.0856 ** | −0.0056 |
| | (0.0463) | (0.0552) | (0.0319) | (0.0332) |
| top1 | 0.0018 | −0.0004 | −0.0017 ** | −0.0006 |
| | (0.0011) | (0.0012) | (0.0008) | (0.0008) |
| rid | 0.3382 | 0.1798 | 0.3043 | −0.1168 |
| | (0.2713) | (0.2876) | (0.2035) | (0.1937) |
| dra | −0.1354 ** | 0.0555 | 0.0037 | −0.0341 * |
| | (0.0430) | (0.0486) | (0.0213) | (0.0199) |
| Year | YES | YES | YES | YES |
| Industry | YES | YES | YES | YES |
| _cons | 5.3711 *** | 5.3509 *** | 5.5301 *** | 5.4607 *** |
| | (0.2391) | (0.2452) | (0.1801) | (0.1678) |
| $N$ | 6537 | 2497 | 10,269 | 4977 |
| $R^2$ | 0.175 | 0.079 | 0.087 | 0.05 |

**Table 8.** *Cont.*

| | (1) ESGlevel | (2) ESGscore | (3) ESGlevel | (4) ESGscore | |
|---|---|---|---|---|---|
| | \multicolumn Panel C: High-pollution industry and low-pollution industry | | | | |
| | High-polluting industry | High-polluting industry | Low-polluting industry | Low-polluting industry | |
| ins | 0.0092 *** | 0.0031 *** | 0.0109 *** | 0.0044 *** | |
| | (0.0006) | (0.0008) | (0.0006) | (0.0005) | |
| age | 0.1022 ** | −0.0203 | 0.1238 ** | 0.0631 * | |
| | (0.0502) | (0.0608) | (0.0455) | (0.0355) | |
| cash | 0.4265 *** | 0.4545 ** | 0.7529 *** | 0.2029 ** | |
| | (0.1207) | (0.1789) | (0.1122) | (0.0915) | |
| lev | −0.0721 | 0.3239 ** | −0.1654 ** | −0.0279 | |
| | (0.0761) | (0.0989) | (0.0749) | (0.0601) | |
| bm | 0.6552 *** | 0.3209 *** | 0.6627 *** | 0.1400 ** | |
| | (0.0629) | (0.0758) | (0.0616) | (0.0477) | |
| growth | 0.0583 | −0.0042 | 0.0281 | −0.0185 | |
| | (0.0390) | (0.0586) | (0.0368) | (0.0326) | |
| top1 | −0.0005 | 0.0011 | 0.0037 *** | −0.0005 | |
| | (0.0009) | (0.0013) | (0.0009) | (0.0008) | |
| rid | 0.2088 | 0.0464 | 0.4742 ** | 0.0465 | |
| | (0.2367) | (0.3188) | (0.2304) | (0.1852) | |
| dra | −0.0959 *** | −0.0439 | −0.0870 ** | −0.0398 * | |
| | (0.0277) | (0.0360) | (0.0267) | (0.0213) | |
| Year | YES | YES | YES | YES | |
| Industry | YES | YES | YES | YES | |
| _cons | 5.3586 *** | 5.4133 *** | 5.1787 *** | 4.9816 *** | |
| | −0.198 | −0.2458 | −0.4397 | −0.179 | |
| *N* | 8340 | 2070 | 8466 | 5404 | |
| $R^2$ | 0.128 | 0.057 | 0.179 | 0.06 | |

The *t*-statistics are presented in the parentheses and the superscripts ***, **, and * denote statistical significance at the 1%, 5%, and 10% levels, respectively.

Secondly, we investigated the moderation effect of different property rights on corporate ESG performance of institutional investors. In panel B, the positive relationship between institutional ownership and ESG performance are significant in both the SOE group and non-SOE group. However, the coefficients of ESGlevel and ESGscore in the SOE group are larger than the coefficients of ESGlevel and ESGscore in the non-SOE group. This means institutional investors in the SOE group can help improve ESG more effectively than the non-SOE group.

Thirdly, we investigated the moderation effect of different industries on corporate ESG performance of institutional investors. In the process of producing products, different industries cause different degrees of pollution and damage to the environment, and as a result, the performance of ESG is also different. We divided the total sample into a high-pollution industry group and a low-pollution industry group. The Ministry of Environmental Protection (MEP) in China issued guidelines on Environmental Information Disclosure for listed companies, stipulating sixteen industries, including coal, metallurgy, chemical, and petrochemical industries, as high-polluting industries. According to this regulation, the enterprises belonging to these sixteen industries are defined as polluting enterprises, and the rest are non-polluting enterprises, and the grouped regression is conducted again. As can be seen from panel C of Table 8, the positive relationship between institutional ownership and ESG performance is significant in both the high-pollution industry group and the low-pollution industry group. However, the coefficients of ESGlevel and ESGscore in the low-pollution group are larger than the coefficients of ESGlevel and

ESGscore in the high-pollution group. It means institutional investors make more efforts to improve ESG in the low-pollution group and the high-pollution group.

The table reports the relationship between institutional ownership and ESG performance in the subcategories of ESG, SOE and non-SOE firms, and high-pollution and low-pollution industry firms. Each regression includes year and firm fixed effects.

## 5. Conclusions

In this study, we found the ratio of institutional ownership had a significantly positive effect on firms' ESG performance. After a series of robustness tests, the results remain unchanged by changing specifications, variable definitions, and using a lagged period. Furthermore, the performance of the environment has been most promoted and the improvement of corporate governance is minimal. The mechanism test suggested that institutional investors can improve ESG performance by actively affecting the personnel changes in management and participating in board voting. According to the heterogeneity test, institutional shareholders have stronger positive effects in SOE firms and low-pollution industry firms.

Theoretically, this paper enriches the literature on the impact of institutional shareholders and the channel of ESG performance improvement. On the one hand, institutional shareholders can use the voice brought by their shareholding to participate in corporate decision making. They can incentivize firms to engage in ESG by management change and board voting. On the other hand, firms can attract the institutional investors as shareholders to improve ESG performance.

Our research has the following implications for investors and policymakers. For institutional investors, they should actively participate in internal firms' governance to express their voice by the shareholders' power. For firms, they can attract institutional investors as shareholders to promote long-term sustainable development, such as ESG performance.

There are several limitations in this paper. Firstly, the paper does not investigate the effect of institutional investors' heterogeneity on firms' ESG performance, such as the ownership background, the pressure on short-term interests, and long-term interests. Secondly, the paper could explore more causal analysis and endogeneity tests to prove the relationship. Further research is needed to overcome these limitations. Therefore, we will search for better robustness tests to enhance our research and collect more heterogeneity information of institutional investors for future research.

**Author Contributions:** Data curation, L.C.; Formal analysis, F.J.; Resources, Y.L.; Writing—original draft, L.H.; Writing—review & editing, B.X. All authors have read and agreed to the published version of the manuscript.

**Funding:** Research Funding of Wuhan Polytechnic University No. 2022RZ005.

**Institutional Review Board Statement:** Not applicable.

**Data Availability Statement:** We obtained the data from the China Stock Market & Accounting Research (CSMAR) Database (https://www.gtarsc.com/, accessed on 3 November 2022) and Wind Database (https://www.wind.com.cn/, accessed on 3 November 2022).

**Conflicts of Interest:** The authors declare no conflict of interest.

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
