# Peer review of "Institutional Shareholders and Firm ESG Performance: Evidence from China"

_sustainability, doi:10.3390/su142214674_

Round 1

Reviewer 1 Report

Dear Authors, Thank you for offering me the opportunity to review this study.

The authors have designed an interesting research project that produces interesting outcomes. The manuscript is well structured, and it is of interest for the readership of the Journal. However, there are some weaknesses that should be corrected before considering the manuscript for publication. They are discussed below.

The study must have a solid theoretical foundation. The authors must rework their general theoretical framework incorporating some classic arguments in the research domain. I did not see any links between this work and the concept of sustainability. But firm ESG performance has a close relationship with sustainability.

You need to add research hypotheses. What do you want to prove as a result of your research?

The manuscript mentioned a gap in the literature (lines 41-42). It is expected by the end of this manuscript I the reader see how the manuscript fill this gap through the results of the research. But I did not see this in the conclusion.

I also suggest the authors to rework the section of conclusion incorporating theoretical discussions. I suggest that the authors reinforce the conclusions by highlighting the value of the research, highlighting the differences from the analyzes of previous scholars, and offering new lines of future research.

I’d be grateful if the authors could provide readers with a summary statistics of the sample.

The manuscript needs a major English editing

Author Response

Response to Reviewer

Point 1:  The study must have a solid theoretical foundation. The authors must rework their general theoretical framework incorporating some classic arguments in the research domain.

Response: Thank you for pointing it out. Following your suggestion, we add the agency theory and stakeholder theory in the arguments in the section of hypothesis. The details are as follows:

In the past decade, sustainable and responsible investments (SRI) have become part of mainstream investing strategies. Some studies pay attention to the relationship between institutional shareholders and corporate ESG performance. There are two opposite views on the role of institution shareholders on ESG performance. On the one hand, the institutional shareholders can improve the ESG performance by monitoring motivation. On the other hand, the institutional shareholders can inhibit the ESG performance by myopia motivation.

According to agency theory, institutional investors, especially mutual funds and pension funds, are active and effective monitors for internal corporate governance (Edmans and Holderness, 2017). Because of their strong ability to collect information, institutional shareholders can effectively monitor the corporate governance and influence the decision-making through the advantages of resources and expertise. Moreover, investment horizons of institutional investors can affect monitoring motivation, which in turn affects various decisions of firms (El Ghoul and Guedhami, 2013; Harford et al., 2018), such as ESG activities.

However, according to stakeholder theory, some prior studies believe the institutional shareholders can inhibit the ESG performance by myopia. The institutional shareholders’ concerns about short-term interests will lead the firm to pursue short-term profits (Graham et al., 2013).  Although the improvement of ESG performance can increase firm value and shareholder wealth in the long term (Lins et al., 2017). The practice of ESG, such as environmental protection and employee welfare, will reduce the earnings and cash in the current term by adding additional expenses and costs (Chen et al, 2017). Furthermore, the disclosure of the improvement in firms’ ESG performance would reduce the market value (Grewal et al., 2018). Therefore, the improvement of ESG performance is at the expense of shareholders’ wealth. Based on this analysis, institutional shareholders will have the incentive to reduce the firms’ ESG performance. 

In the total, based on the prior studies, this paper examines the relationship between institutional investors and ESG performance of A-shares list firms in China. We propose the following competitive hypothesis.

H1: Institutional investors can improve the ESG performance in A-shares list firms.  

H2: Institutional investors can inhibit the ESG performance in A-shares list firms.

Point 2: I did not see any links between this work and the concept of sustainability. But firm ESG performance has a close relationship with sustainability.

Response:  Thank you for pointing it out. Following your suggestion, we explain the relationship between firms’ ESG performance and sustainability in the section of introduction. The details are as follows:

The growing governments value the sustainable and responsible impact in the development of economic and encourage the financial institution care the social responsibility in the investment. In recent years, the situation of climate change, energy depletion and environment pollution seriously affect the sustainable development in the world. In 2006, the United Nations set up the Principles and Responsible Investment (PRI) to encourage the financial institution members to commit to responsible investment.   Under the social demand for responsible investments, a growing number of institutions join in PRI and take ESG (Environment, Social Responsibility, Corporate Governance) into consideration. As shown in figure 1, 4670 financial institutions are the member of PRI at the end of 2021. Firms’ ESG performance means the firms’ pursuit of maximization of social interests. ESG practice is the channel for firms to achieve sustainable development goals.

Point 3:   You need to add research hypotheses. What do you want to prove as a result of your research?

Response:  Thank you for pointing it out. Following your suggestion, we add the proof to the hypotheses in our research. The details are as follows:

In the past decade, sustainable and responsible investments (SRI) have become part of mainstream investing strategies. Some studies pay attention to the relationship between institutional shareholders and corporate ESG performance. There are two opposite views on the role of institution shareholders on ESG performance. On the one hand, the institutional shareholders can improve the ESG performance by monitoring motivation. On the other hand, the institutional shareholders can inhibit the ESG performance by myopia motivation.

According to agency theory, institutional investors, especially mutual funds and pension funds, are active and effective monitors for internal corporate governance (Edmans and Holderness, 2017). Because of their strong ability to collect information, institutional shareholders can effectively monitor the corporate governance and influence the decision-making through the advantages of resources and expertise. Moreover, investment horizons of institutional investors can affect monitoring motivation, which in turn affects various decisions of firms (El Ghoul and Guedhami, 2013; Harford et al., 2018), such as ESG activities.

There are three main motives for institutional investors to care about the ESG performance. First, the beliefs of institutional investors in social responsibility have important impact on the involvement in SRI (Williams, 2007). The questionnaires released by Zwaan (2015) shows the sovereign wealth funds, pension funds and NGO funds are more interested in firms with higher ESG performance.

Second, the public’s preference for ESG will encourage institutional investors to invest in firms with high ESG level. More and more individual investors will consider environmental and social impacts when making investment decisions. They are willing to invest in socially responsible firms with a high financial premium (Flammer et al., 2010). Social preferences are more effective than financial motives in the explanation of SRI (Alexander et al., 2017).

Third, institutional investors can use SRI in portfolios for risk management. They believe that environmental risks have financial implications for their portfolio. Especially in the long-term investment, it’s a better approach for participating in the environmental risk management rather than divestment (Krueger et al., 2020). Jose Azar et al (2021) find that Big Three (i.e., BlackRock, Vanguard, and State Street Global Advisors) focus their portfolio on the firms with high CO2 emissions and make effort on reducing the CO2 emissions from these firms. Philipp et al. (2020) find that companies with better ESG performance tend to have investors with longer investment horizons.

However, according to stakeholder theory, some prior studies believe the institutional shareholders can inhibit the ESG performance by myopia. The institutional shareholders’ concerns about short-term interests will lead the firm to pursue short-term profits (Graham et al., 2013).  Although the improvement of ESG performance can increase firm value and shareholder wealth in the long term (Lins et al., 2017). The practice of ESG, such as environmental protection and employee welfare, will reduce the earnings and cash in the current term by adding additional expenses and costs (Chen et al, 2017). Furthermore, the disclosure of the improvement in firms’ ESG performance would reduce the market value (Grewal et al., 2018). Therefore, the improvement of ESG performance is at the expense of shareholders’ wealth. Based on this analysis, institutional shareholders will have the incentive to reduce the firms’ ESG performance. 

In the total, based on the prior studies, this paper examines the relationship between institutional investors and ESG performance of A-shares list firms in China. We propose the following competitive hypothesis.

H1: Institutional investors can improve the ESG performance in A-shares list firms.  

H2: Institutional investors can inhibit the ESG performance in A-shares list firms.

Point 4: The manuscript mentioned a gap in the literature (lines 41-42). It is expected by the end of this manuscript I the reader see how the manuscript fill this gap through the results of the research. But I did not see this in the conclusion.

Response:  Thank you for pointing it out. Following your suggestion, we add the gap in the literature in the section of introduction. The details are as follows:

The social responsible investment has become a common phenomenon in China’s capital market and the institutional shareholders become more concerned about firms’ ESG performance in the portfolio. It is worthy of attention from the academic community and the industry. Although some prior research show that institutional ownership is positively related to firm’s ESG in the USA and European financial market, just a few studies focus on the relationship between the institution shareholders and firms’ ESG performance in China. Due to the weak social network, QFII can only affect the ESG performance in non-state-owned firm significantly (McGuinness et al., 2017), while the state-owned institutional investors pay attention to social responsibility, such as targeted poverty reduction (Lin et al., 2015). Allen et al.(2016)find that institutional shareholders can drive CSR performance in firms with low financing constraints.

Our study makes three important contributions to the literature on institutional investors, corporate ESG and socially responsible investment. First, our study contributes to the growing literature on corporate ESG.  A lot of recent studies have focused on firm-level characteristics, shareholders characteristics or observable managerial characteristics to explain the variation in firms’ ESG (Appel et al., 2016). With the perspective of institutional shareholders, we find the link between institutional ownership and ESG performance in Chinese financial market in OLS model, order logistic model and tobit model.

Second, we contribute to the literature on theimpact of institutional investors on corporate governance. Different from the previous literature mostly focusing on the institutional background (McGuinness et al., 2017; Lin et al., 2015), we explore the channel on the improvement of ESG performance derived by institutional shareholders. Our research shows the institutional shareholders make real effort to promote ESG performance by affecting the personnel changes in management and participating in board proposals.

Third, different from the previous literatures taking the ESG performance of Chinese firms as a whole (Feng et al., 2022; Hao et al., 2022), we further analyze institutional investors’ impact on the three subcategories of ESG and find the performance of environment has been promoted the most.

Point 5:  I also suggest the authors to rework the section of conclusion incorporating theoretical discussions. I suggest that the authors reinforce the conclusions by highlighting the value of the research, highlighting the differences from the analyzes of previous scholars, and offering new lines of future research.

Response:  Thank you for pointing it out. Following your suggestion, we have reworked the section of conclusion. We reinforce the conclusion by highlighting the value of the research and offering new lines of future research. But we decide to highlight the differences from the analyzes of previous scholars in the section of introduction, you can find the details in the response of point 5. The details of conclusion are as follows:

In this study, we find the ratio of institutional ownership has a significantly positive effect on firms’ ESG performance. After a series of robustness tests, the results remain unchanged by changing specifications, variable definitions and using lagged period. Furthermore, the performance of environment has been promoted the most and the improvement of corporate governance is minimal. The mechanism test suggests that institutional investors can improve ESG performance by actively affecting the personnel changes in management and participating in board voting. According to the heterogeneity test, institutional shareholders have stronger positively effect in SOE firms and low-pollution industry firms.

Theoretically, this paper enriches the literature on the impact of institutional shareholders and the channel of ESG performance improvement. On the one hand, institutional shareholders can use the voice brought by their shareholding to participate in corporate decision-making. They can incentivize firms to engage in ESG by management change and board voting. On the other hand, firms can attract the institutional investors as shareholders to improve ESG performance.

Our research has the following implications for investors and policymakers. For institutional investors, they should actively participate in internal firms’ governance to express your voice by the shareholders’ power. For firms, they can attract institutional investor as shareholders to promote long-term sustainable development, such as ESG performance.

There are several limitations in this paper. First, the paper doesn’t investigate the effect of institutional investors’ heterogeneity on firms’ ESG performance, such as the ownership background, the pressure on short -term and long-term interests. Second, the paper can explore more causal analysis and endogeneity test to prove the relationship. Further researches are needed to overcome the limitations.

Point 6:  I’d be grateful if the authors could provide readers with a summary statistics of the sample.

Response:  Thank you for pointing it out. Following your suggestion, we present the summary statistics of the sample (table 3) in the section of empirical results and analysis.

Table 3. Summary Statistics.

The table presents summary characteristics (mean) for the sample firms. The sample comprises 16810 firm-year observations during the 2013 to 2020 period. The appendix provides detailed descriptions of the variables.

variable

N

max

min

mean

p50

ESGlevel

16810

9

1

6.526

6

ESGscore

7477

8.100

4.730

6.099

6.020

ins

16810

83.78

0.491

39.87

41.29

age

16810

3.526

2.398

3.032

3.045

cash

16810

0.536

0.0130

0.149

0.120

lev

16810

0.844

0.0770

0.426

0.418

roa

16810

0.177

-0.145

0.0430

0.0390

size

16810

25.68

20.22

22.29

22.14

bm

16810

1.121

0.153

0.618

0.610

growth

16810

1.367

-0.434

0.144

0.101

top1

16810

68.00

10.52

34.35

32.55

rid

16810

0.500

0.333

0.373

0.364

Table 3 shows the descriptive statistics for all variables in the present study. The final sample consists of 16810 firm-year observations. The statistical results show that the average value of the ESGlevel is 6.526, the max value is 9 and min value is 1. The average value of the ESGscore is 6.099, the max value is 8.1 and the min value is 4.73. According to the distribution of ESGscore and ESGlevel, we can get a large gap in ESG performance among our observations.

The average value of the ratio of institutional ownership is 39.87%, the max value is 83.78%. This shows institutional investors can have an important influence on the company's decision-making in some firms.

Point 7: The manuscript needs a major English editing

Response:  Thank you for pointing it out. Following your suggestion, we rewrite the manuscript sentence by sentence and correct the grammar mistakes.

Reviewer 2 Report

Thank you for inviting me to review this manuscript, titled “Institutional shareholders and firm ESG performance: evidence from China”. The research topic proposed by the authors is current.

Please find my detailed comments below:

Regarding the abstract, I suggest that the authors outline the methodology. The authors should develop the discussion section, deepening the analysis by identifying the causes that generate the results, and signaling the outcomes’ meaning, importance, and applicability.Furthermore, several aspects should be included in the debate: Could ESG performance have a substitution effect on corporate governance?; the relevance of the study - Does China differ significantly in terms of institutional background? Can results be applied to other systems? No reference is made to whether or not the results are in line with previous research. I suggest that the conclusions should be concreted by associating with previous studies in the literature.

Conclusions cover several good points, not including the limitation of the research and recommendations for future research.

I would suggest a proofreading (line 118, 119, 215). 

Thank you for this interesting paper.

Author Response

Response to Reviewer

Point 1:  Regarding the abstract, I suggest that the authors outline the methodology. 

Response: Thank you for pointing it out. Following your suggestion, we rewrite the abstract and outline the methodology in our research. The details are as follows:

It is a noteworthy phenomenon that institutional investors care more about the ESG performance of the firms in their portfolios in China. Exploring the role of institutional shareholders in firms’ ESG performance is vital for corporate sustainable growth. Using a sample of public listed firms from 2013 to 2020 in China, we find that firms with higher institutional ownership have better ESG performance in OLS model, order logistic model and tobit model, especially in the environmental (E) aspect. The positive effect of institutional investor on ESG performance is more pronounced in SOE firms, and firms in low-pollution industry. Furthermore, mechanism test suggests that institutional shareholders can incentivize firms to engage in ESG by affecting management change and board voting.

Point 2: The authors should develop the discussion section, deepening the analysis by identifying the causes that generate the results, and signaling the outcomes’ meaning, importance, and applicability.

Response: Thank you for pointing it out. Following your suggestion, we analyze the causes that generate the results in the hypothesis section and we highlight the outcomes’ meaning, importance, and applicability in the conclusion section.

The details of causes that generate the results are as following:

In the past decade, sustainable and responsible investments (SRI) have become part of mainstream investing strategies. Some studies pay attention to the relationship between institutional shareholders and corporate ESG performance. There are two opposite views on the role of institution shareholders on ESG performance. On the one hand, the institutional shareholders can improve the ESG performance by monitoring motivation. On the other hand, the institutional shareholders can inhibit the ESG performance by myopia motivation.

According to agency theory, institutional investors, especially mutual funds and pension funds, are active and effective monitors for internal corporate governance (Edmans and Holderness, 2017). Because of their strong ability to collect information, institutional shareholders can effectively monitor the corporate governance and influence the decision-making through the advantages of resources and expertise. Moreover, investment horizons of institutional investors can affect monitoring motivation, which in turn affects various decisions of firms (El Ghoul and Guedhami, 2013; Harford et al., 2018), such as ESG activities.

There are three main motives for institutional investors to care about the ESG performance. First, the beliefs of institutional investors in social responsibility have important impact on the involvement in SRI (Williams, 2007). The questionnaires released by Zwaan (2015) shows the sovereign wealth funds, pension funds and NGO funds are more interested in firms with higher ESG performance.

Second, the public’s preference for ESG will encourage institutional investors to invest in firms with high ESG level. More and more individual investors will consider environmental and social impacts when making investment decisions. They are willing to invest in socially responsible firms with a high financial premium (Flammer et al., 2010). Social preferences are more effective than financial motives in the explanation of SRI (Alexander et al., 2017).

Third, institutional investors can use SRI in portfolios for risk management. They believe that environmental risks have financial implications for their portfolio. Especially in the long-term investment, it’s a better approach for participating in the environmental risk management rather than divestment (Krueger et al., 2020). Jose Azar et al (2021) find that Big Three (i.e., BlackRock, Vanguard, and State Street Global Advisors) focus their portfolio on the firms with high CO2 emissions and make effort on reducing the CO2 emissions from these firms. Philipp et al. (2020) find that companies with better ESG performance tend to have investors with longer investment horizons.

However, according to stakeholder theory, some prior studies believe the institutional shareholders can inhibit the ESG performance by myopia. The institutional shareholders’ concerns about short-term interests will lead the firm to pursue short-term profits (Graham et al., 2013).  Although the improvement of ESG performance can increase firm value and shareholder wealth in the long term (Lins et al., 2017). The practice of ESG, such as environmental protection and employee welfare, will reduce the earnings and cash in the current term by adding additional expenses and costs (Chen et al, 2017). Furthermore, the disclosure of the improvement in firms’ ESG performance would reduce the market value (Grewal et al., 2018). Therefore, the improvement of ESG performance is at the expense of shareholders’ wealth. Based on this analysis, institutional shareholders will have the incentive to reduce the firms’ ESG performance. 

In the total, based on the prior studies, this paper examines the relationship between institutional investors and ESG performance of A-shares list firms in China. We propose the following competitive hypothesis.

H1: Institutional investors can improve the ESG performance in A-shares list firms.  

H2: Institutional investors can inhibit the ESG performance in A-shares list firms.

The details of the outcomes’ meaning, importance, and applicability are as following:

In this study, we find the ratio of institutional ownership has a significantly positive effect on firms’ ESG performance. After a series of robustness tests, the results remain unchanged by changing specifications, variable definitions and using lagged period. Furthermore, the performance of environment has been promoted the most and the improvement of corporate governance is minimal. The mechanism test suggests that institutional investors can improve ESG performance by actively affecting the personnel changes in management and participating in board voting. According to the heterogeneity test, institutional shareholders have stronger positively effect in SOE firms and low-pollution industry firms.

Theoretically, this paper enriches the literature on the impact of institutional shareholders and the channel of ESG performance improvement. On the one hand, institutional shareholders can use the voice brought by their shareholding to participate in corporate decision-making. They can incentivize firms to engage in ESG by management change and board voting. On the other hand, firms can attract the institutional investors as shareholders to improve ESG performance.

Our research has the following implications for investors and policymakers. For institutional investors, they should actively participate in internal firms’ governance to express your voice by the shareholders’ power. For firms, they can attract institutional investor as shareholders to promote long-term sustainable development, such as ESG performance.

Point 3: Furthermore, several aspects should be included in the debate: Could ESG performance have a substitution effect on corporate governance?

Response: Thank you for pointing it out. Following your suggestion, we analyze the substitution effect of ESG performance on corporate governance in the addition test in the section of empirical results and analysis. The details are as follows:

To better understand what aspect of ESG issues are most affected by institutional shareholders, we extend the baseline specification to separately study the effect of shareholders on the three different dimensions of ESG activities. In the panel A of table 8, we can see the different relationship between E(environment), S(Society), G(Governance) and institutional ownership in OLS regression of column (1), (2), (3) and tobit regression in column (4), (5), (6). Both in OLS regression and tobit regression models, we can get that the coefficients of E are the largest, then followed by S at 1% significant level. The coefficients of G are the smallest at 10% significant level. We can get that institutional investors pay most attention to the environmental protection performance of companies, and they will actively help companies improve environmental management, increase environmental information disclosure and reduce negative environmental events. Such as reducing CO2 emissions, increasing the use of renewable energy, community contribution, product liability and so on.

Point 4:   The relevance of the study - Does China differ significantly in terms of institutional background? Can results be applied to other systems? No reference is made to whether or not the results are in line with previous research. 

Response: Thank you for pointing it out. Following your suggestion, we explain the institutional background in China in the introduction section and the results can be applied to other systems in the conclusion section.

The details of Chinese institutional background are as follows: In the Chinese financial market, the institutional investments include sovereign wealth fund, mutual fund, securities, insurance fund, social security fund, annuity, privately offered fund and QFII (Qualified foreign institutional investors). Different from some western countries, most sovereign wealth funds, insurance funds and social security funds are guided by Chinese government or state capital (Hong and Xu., 2019). Meanwhile, the protection environment for institutional investors belongs to non-state-owned capital is relatively weak compared to some western countries (Feng et al., 2022). 

The details of the results be applied to other systems: Our research has the following implications for investors and policymakers. For institutional investors, they should actively participate in internal firms’ governance to express their voice by the shareholders’ power. For firms, they can attract institutional investor as shareholders to promote long-term sustainable development, such as ESG performance.

Point 5:  I suggest that the conclusions should be concreted by associating with previous studies in the literature.

Response: Thank you for pointing it out. Following your suggestion, we explain the connection between our research and previous studies in the introduction section and conclusion section.  The details are as following:

Introduction: Our study makes three important contributions to the literature on institutional investors, corporate ESG and socially responsible investment. First, our study contributes to the growing literature on corporate ESG. A lot of recent studies have focused on firm-level characteristics, shareholders characteristics or observable managerial characteristics to explain the variation in firms’ ESG (Appel et al., 2016). With the perspective of institutional shareholders, we find the link between institutional ownership and ESG performance in Chinese financial market in OLS model, order logistic model and tobit model.

Second, we contribute to the literature on the impact of institutional investors on corporate governance. Different from the previous literature mostly focusing on the institutional background (Mc Guinness et al., 2017; Lin et al., 2015), we explore the channel on the improvement of ESG performance derived by institutional shareholders. Our research shows the institutional shareholders make real effort to promote ESG performance by affecting the personnel changes in management and participating in board proposals.

Third, different from the previous literatures taking the ESG performance of Chinese firms as a whole (Feng et al., 2022; Hao et al., 2022), we further analyze institutional investors’ impact on the three subcategories of ESG and find the performance of environment has been promoted the most.

Conclusion: Theoretically, this paper enriches the literature on the impact of institutional shareholders and the channel of ESG performance improvement. On the one hand, institutional shareholders can use the voice brought by their shareholding to participate in corporate decision-making. They can incentivize firms to engage in ESG by management change and board voting. On the other hand, firms can attract the institutional investors as shareholders to improve ESG performance.

Point 6: Conclusions cover several good points, not including the limitation of the research and recommendations for future research.

Response: Thank you for pointing it out. Following your suggestion, we present the limitation of our research and recommendations for future research in the conclusion section. The details are as following:

There are several limitations in this paper. First, the paper doesn’t investigate the effect of institutional investors’ heterogeneity on firms’ ESG performance, such as the ownership background, the pressure on short -term and long-term interests. Second, the paper can explore more causal analysis and endogeneity test to prove the relationship. Further researches are needed to overcome the limitations. Therefore, we will search better robustness tests to enhance our research and collect more heterogeneity information of institution investors for future research.

Point 7:  I would suggest a proofreading (line 118, 119, 215)

Response: Thank you for pointing it out. Following your suggestion, we proofreading of line 118, 119, 215. The details are as following:

The proofreading of 118 and 119: The female board members will pay more attention to social issues and environment issues and the firms will have higher ESG performance (McGuiness et al., 2017). In the USA, the ESG ratings can be improved 9.1% if the CEO has a daughter (Cronqvist and Yu., 2019).  The CEOs with military experience care more about projects with higher ESG performance (Benmelech and Frydman., 2015).

The proofreading of 215: Following Dyck et al., (2018), we estimate the following baseline model to investigate the relationship between shares held by institutional shareholders and ESG performance:

Round 2

Reviewer 1 Report

Thanks to the authors for the excellent work on finalizing the manuscript.

Now it can be published.

I wish you success in your further research.

Reviewer 2 Report

The authors have improved the content of the document. All suggestions and recommendations were dealt with in a detailed and correct manner.

Kind regards,